# Analysis of Recent Bio-/Nanotechnologies for Coronavirus Diagnosis and Therapy

**DOI:** 10.3390/s21041485

**Published:** 2021-02-20

**Authors:** Amina Rhouati, Ahlem Teniou, Mihaela Badea, Jean Louis Marty

**Affiliations:** 1Bioengineering Laboratory, Higher National School of Biotechnology, Constantine 25016, Algeria; teniouahlem97@gmail.com; 2Faculty of Medicine, Transilvania University of Brasov, 500039 Brasov, Romania; mihaela.badea@unitbv.ro; 3Laboratoire BAE, Université de Perpignan Via domitia, 66860 Perpignan, France

**Keywords:** coronavirus, diagnosis, treatment, biomolecules, nanoparticles, biosensing, drug delivery

## Abstract

Despite barrier measures and physical distancing tailored by the populations worldwide, coronavirus continues to spread causing severe health and social-economic problems. Therefore, researchers are focusing on developing efficient detection and therapeutic platforms for SARS-CoV2. In this context, various biotechnologies, based on novel molecules targeting the virus with high specificity and affinity, have been described. In parallel, new approaches exploring nanotechnology have been proposed for enhancing treatments and diagnosis. We discuss in the first part of this review paper, the different biosensing and rapid tests based on antibodies, nucleic acids and peptide probes described since the beginning of the pandemic. Furthermore, given their numerous advantages, the contribution of nanotechnologies is also highlighted.

## 1. Introduction

To date, six human coronaviruses have been identified: α-coronaviruses (HCoVs-NL63, HCoVs-229E), β-coronaviruses (HCoVs-OC43, HCoVs-HKU1), severe acute respiratory syndrome-CoV (SARS-CoV), and Middle East respiratory syndrome-CoV (MERS-CoV) [1]. After the SARS-CoV-1 epidemic, the world is living a new threat to human health since December 2019—the SARS-CoV-2 or the COVID-19 pandemic. The emergence of the novel coronavirus is associated with an atypical pneumonia that has led to 90,176,569 infections and 1,936,617 deaths worldwide, as of 10 January 2021. Structurally, SARS-CoV-2 is an enveloped RNA(Ribonucleic acid) virus comprising a spike protein (S), a hemagglutinin-esterase dimer (HE), a membrane glycoprotein (M), an envelope protein (E), and a nucleocapsid protein (N) [2]. It has been demonstrated that the mechanism of the viral infection requires angiotensin-converting enzyme 2 (ACE2) binding to the protein S with high affinity. Highly expressed in the endothelial cells of the cardiovascular system and kidneys, this human receptor is used by the virus as an entry to invade target cells [3].

Different measures have been imposed by authorities, including wearing masks and physical distancing, aiming to stop the virus spread. However, the adopted procedures are not sufficient, and mass screening of coronavirus is more than a necessity. At present, real-time RT-PCR (Reverse Transcriptase-Polymerase Chain Reaction) is the most used technique for COVID-19 testing. It is based on specific primers targeting sequences encoding the proteins E, N, S, ORF1b(Open Reading Frame 1b), and RdRp(RNA-dependant RNA polymerase)or a combination of different targets [4,5]. The main advantage of this technique remains in the sensitive detection of the virus at the onset of infection [6]. In parallel, serological diagnosis based on measuring the anti-viral antibodies in the blood, is employed to monitor the immune response developed by patients [7,8]. Recently, rapid tests based on targeting the antigenic protein of the virus by using immuno-chromatography are also available. Despite the short time of response, antigen tests suffer from poor sensitivity [9].

Apart from familiar technologies, there have been several reports describing the identification of new biorecognition elements that have been employed in different biosensing schemes [10,11]. In addition, nanomaterials have been integrated in certain detection strategies for different roles, including signal amplification and biomolecule immobilization [12,13]. Therefore, the first part of this review will be dedicated to discussing the different biosensors developed since the outbreak of the COVID-19 crisis. Then, an overview of the innovative therapeutic platforms will be given by focusing on the contribution of nanotechnology to coronavirus treatment.

## 2. Novel Biosensing Strategies Used for COVID-19 Diagnosis

Several research groups are working on the development of rapid and sensitive assays and biosensors to overcome the limitations of traditional techniques used in coronaviruses detection. These techniques are mainly based on antibodies and nucleic acids including aptamers, clustered regularly interspaced short palindromic repeats (CRISPR), and antisense oligonucleotides.

### 2.1. Immunochemical Techniques

Currently, immunoassays are the most popular diagnostic tools available in the market and used in medical structures. Basically, these methods use antibodies as bioreceptors targeting capsid proteins or whole viruses. In serological testing, capsid proteins are used as viral antigens to bind the immunoglobulins generated by the patient against the pathogen. Antibodies are usually obtained from animal immunization with N, S, or E protein or from the blood samples of patients who are infected [14]. In addition to the commercialized ELISA kits and rapid tests, several research reports have described novel immunoassays and immunosensors for coronavirus detection. We discuss in this part the principle of these methods as well as the most important results. 

#### 2.1.1. Electrochemical Immunosensors

Electrochemical immunosensing is based on the conversion of the antibody–antigen interaction into electrochemical signals. In contrast to traditional immunoassays, innovative transducers allow a highly sensitive quantification of the targeted antigen/antibody in different ways [15]. It should be noted that sensitivity plays a crucial role in viruses detection; lowering the detection limit allows the early diagnosis of the infection before the first symptoms [16]. In addition to their sensitivity and robustness, electrochemical devices can be miniaturized, providing an immediate response without referring to specialized labs and qualified staff [17]. For instance, Fabiani et al. employed carbon-black-based screen-printed electrodes as transducers and magnetic beads as an immobilization support for the immunological chain comprising anti-IgG (Immunoglobulin G) bound to the anti-S and anti-N antibodies. The principle is based on the specific recognition of the anti-S and/or anti-N antibodies with the virus present in the sample. After the antibody–antigen interaction, polyclonal anti-S and anti-N antibodies are added, followed by an alkaline-phosphatase-labeled secondary antibody. Finally, the conversion of the enzymatic substrate to the electro-active by-product 1-naphtol was monitored by differentialpulse voltammetry (DPV) using a portable potensiostat connected to a computer. The immunosensor analytical performances were confirmed in standard solutions of S and N proteins and untreated saliva. Moreover, experiments using cultured virus at biosafety level 3 and in saliva clinical samples were carried out by comparing the data using the nasopharyngeal swab specimens tested with RT-PCR. Despite its high sensitivity for SARS-CoV-2 and selectivity toward H1N1 influenza virus, the reported immunosensor requires several steps of construction and antibodies, thusincreasing the complexity and cost of the method [18]. Later, a simple and low-cost electrochemical immunosensing strategy was reported for the simultaneous measurement of the nucleocapsid protein, IgG and IgM (Immunoglobulin M) specific to the S1 protein, as well as C-reactive protein (CRP) levels. These biomarkers provide information about the infection, immune response, and disease severity. The telemedicine Rapidplex platform is composed of four graphene electrodes fabricated by laser engraving [19]. Electrodes are designed based on the sandwich/double sandwich for Igsand indirect immunosensing for N and CRP (Figure 1). The effectiveness of each electrode was monitored amperometrically in blood and saliva samples. Excellent results were obtained in terms of sensitivity, selectivity toward SARS-CoV-1 and MERS-CoV, as well as reproducibility and stability [20]. In brief, this innovative platform should meet the high requirement for modern diagnosis tools allowing accurate, simple and rapid monitoring of SARS-CoV2 infection. In a recent study, Zourob’s group developed cotton tipped electrochemical immunosensor for the nucleocapsid protein detection. The novelty of the described strategy is in combing the sample collection and detection tools in a single platform by coating screen-printed electrodes with absorbing cotton padding. The detection principle was based on the competition between the immobilized antigen on screen printed electrodes and that present in nasal samples, to bind N protein antibody in solution. This platform would avoid sample pre-treatment and transfer which are indispensable in the other diagnosis approaches [21].

#### 2.1.2. Field-Effect Transistor-Based Immunosensors

Field-effect transistor(FET)-based biosensing is characterized by many advantages, including sensitivity, fast response, and low-noise detection [22]. Moreover, FET-based biosensing does not require labeling of bioreceptors and targets, making the strategy simpler and cheaper. Therefore, this strategy represents great potential in point-of-care testing and clinical diagnosis of viral infections. Aiming to detect the new coronavirus with high sensitivity, Zhang et al. developed a graphene-FET immunosensor for electrical probing of the spike protein S1 [23]. Graphene is widely used in biosensing platforms because of its remarkable conductivity, carrier mobility, and large surface area [24]. In this work, the subunit S1 was targeted because it is less conserved than N and S2 proteins in SARS-CoV [25]. Furthermore, the S1 protein contains a receptor-binding domain (RBD), which interacts with the ACE2 receptor [26]. In brief, the reported immunosensor was constructed by functionalizing the graphene surface with the S1-specific antibody or ACE2. The formation of the complex antibody/ACE-S1 subunit directly alters the conductance/resistance via field effect, resulting in a measurable electrical signal. The developed platform could provide an alternative platform for early screening and diagnosis. It would be also useful in rational design of neutralizing antibodies and docking methods as well as developing vaccines, prophylactics and therapeutics to combat COVID-19 [23]. Despite the high sensitivity of the reported immunosensor, no results of clinical sample application were reported. In another study, a highly sensitive FET-based immunosensor was constructed and successfully applied for spike protein detection in cultured virus and nasopharyngeal swab specimens from COVID-19 patients [11].

#### 2.1.3. Optical Immunosensors

Besides electrochemical methods, optical biosensors have been widely used in clinical diagnosis and drug discovery due to their analytical performance in detecting biological systems [27]. Different approaches have been reported for optical immunosensing of the novel coronavirus, including surface plasmon resonance (SPR), chemiluminescence, and colorimetry.

Surface plasmon resonance-based biosensors are based on the monitoring of refractive index variations after the formation of the target–receptor complex on the sensing surface. Plasmonic biosensing offers many advantages, including label-free and real-time monitoring [28]. Yong’s group described a sandwich plasmonic immunosensor to detect a spike protein in serum samples. The principle of the assay is based on functionalizing a gold nanosheet with the protein S specific antibody by conjugating gold nanorods to the same antibody for sensitivity enhancement. Then, samples containing the antigen were allowed to flow as an analyte. This work was assisted by a simulation technique to study the effect of ARs and arrangements of the gold nanorods [29].

Besides label-free sensors, optical sensors, based on chemiluminescent and colorimetric labeling, have been reported for COVID-19 diagnosis. First, Cai et al. developed a chemiluminescent enzymatic serological immunoassay based on a peptide from the S protein. The latter was screened from peptide antigens synthesized from the genomic sequence from GenBank (NC_045512.1). Streptavidin-functionalized magnetic beads were used as an immobilization support for the biotinylated peptide. The binding between IgM and IgG present in serum samples and the magnetic beads was carried out in solution. After binding and washing, the conjugate was reacted with the substrate, generating a luminescent signal, which was measured by a luminometer. In this work, no clinical application was carried out [30]. Another research group reported, subsequently, the development and clinical application of a colorimetric qualitative lateral flow immunoassay for simultaneous detection of IgM and IgG in blood. The rapid test is composed of anti-IgG and anti-IgM antibodies stripped on two separate test lines. In parallel, a conjugate of viral antigen-AuNPs is sprayed on conjugation pads to generate a colorimetric signal. The presence of IgG and/or IgM antibodies is indicated by a red/purple line. The rapid test was applied on 397 PCR-confirmed patients and 128 negative patients at eight different clinical sites, showing an overall testing sensitivity of 88.66% and a specificity of 90.63% [31].

Despite the importance of IgGs in detecting previous COVID-19 infection and the association of IgM to recent contamination, the role of secretory immunoglobulins IgA should not be neglected. IgAs(Immunoglobulin A)play a key role in the immune exclusion process limiting the access of microorganisms to the mucosal barriers. It has been demonstrated that elevated total IgA and IgA-aPL(IgA-antiphospholipid) are significantly associated with severe COVID-19 infection [32,33]. Roda et al. developed a dual optical immunosensing platform for IgA detection in saliva by using the nucleocapsid protein as a receptor. The anti-human IgA was labeled with AuNPs for colorimetric detection, while HRP(HorsradishPeroxidase)was used to generate chemiluminescence after reaction with the luminol substrate. IgA present in serum/saliva is recognized by the immobilized N protein, and the secondary antibody captures the formed complex, thus generating a colorimetric/chemiluminescent signal. For colorimetry, a smartphone camera was used to capture the colored strip, while the light emitted by the chemiluminescence reaction (luminol-HRP) was measured by a cooled CCD (Charge Coupled Device) in contact imaging mode, with the data reported in relative light units. In contrast to the previously discussed LFIA (Lateral Flow Immunoassay), the one described in this report provides a qualitative and quantitative analysis. The use of this rapid test would assist studies on the role of IgA in the SARS-CoV-2 [34]. In addition to immunoglobulins, rapid tests based on LFIA have been also reported for capsid protein detection. For instance, Diao et al. developed a fluorescent immunoassay to detect a nucleocapsid protein in nasopharyngeal swab samples and urine within 10 min [35].

The different immunosensing strategies described above are summarized in Table 1 and Table 2.

### 2.2. Nucleic Acid-Based Techniques

As coronaviruses are positive single-stranded RNA viruses, their detection is usually based on nucleic acid techniques, mainly real-time PCR. However, this technology requires expensive reagents and equipment, in addition to qualified manipulators. Therefore, researchers are focusing on new techniques based on clustered regularly interspaced short palindromic repeats (CRISPR), aptamers, and nucleic acidhybridization to reduce the dependence on RT-PCR.

#### 2.2.1. Clustered Regularly Interspaced Short Palindromic Repeats (CRISPR)-Based Biosensing

After the identification of CRISPR-Cas systems targeting RNA, numerous studies have shown the potential use of this tool in the detection of different viruses such as dengue and Zika viruses [50]. Cas13 effector proteins are able to target RNA, with two RNase activities: gRNA maturation and both cis- and trans-RNA target cleavage [51]. They hold, thus, great promise for SARS-CoV-2 diagnosis and treatment. CRISPR-based platforms provide sensitive, rapid, and low-cost virus detection. Moreover, CRISPR-based diagnosis is usually based on isothermal amplification techniques that can be performed without thermocycler (recombinase polymerase amplification (RPA) and loop-mediated isothermal amplification (LAMP)). Several platforms have been applied for COVID-19 detection: specific high-sensitivity enzymatic reporter unlocking (SHERLOCK), HOLMES (one-hour low-cost multipurpose highly efficient system), DNA (Desoxyribonucleic acid), endonuclease-targeted CRISPR trans reporter (DETECTR), CAS-EXPAR(EXPonantial Amplification Reaction), NASBACC (nucleic acid sequences-based amplification CRISPR Cleavage), STOPCovid, ctPCR (cycle threshold PCR), and AIOD-CRISPR(All in one dual crispr). These platforms have been well discussed in a critical review by Vatankhah et al. [52].

The first designed and most used platform, called SHERLOCK showed a successful application for identifying cell-free tumor DNA mutations, genotyping human DNA, detectingdengue and Zika viruses, and distinguishing pathogenic bacteria [52]. Several studies have also demonstrated the potential application of this platform for COVID-19 detection [53]. SHERLOCK is simply based on Cas13a activation by the binding and cleavage of the target RNA. As a result, the nontarget molecules will be degraded including the quenched fluorescent reporter molecules. The released signal allows thus detection of the specific RNA after degradation of the non-targeted one [50]. Based on this principle, Zhang et al reported the first CRISPR-Cas13-based technique for COVID-19 detection by targeting S and ORF1ab protein genes. The presence of coronavirus in the sample activates the cas13 cleavage system by generating a measurable fluorescent signal. The developed technique allows the quantification of the coronavirus target genes in the range (20–200 aM) corresponding to (10–100 copies/µL). The whole test can be performed using a dipstick in less than one hour [54]. CRISPR-Cas13 technology can be thus considered as an excellent alternative to qRT-PCR for coronavirus diagnosis. In addition, the FDA (Food and Drug Administration) authorized the use of the Sherlock CRISPR SARS-CoV-2 Kit proposed by Sherlock BioSciencesCompany in May, 2020 [55] Subsequently, the clinical validation of this test has been reported by Patchsung et al in 154 nasopharyngeal and throat swab samples collected at Siriraj Hospital, Thailand [56].

By combining SHERLOCK with the HUDSON protocol; a process for lysis of viral particles and inactivation of DNases and RNases in body fluids with heat and chemical treatment, no prior RNA extraction is needed [52]. In a recent study, HUDSON was improved to allow virus inactivation within 10 minthusproviding a simple, rapid and single step Cas-13 detection tool of COVID-19. The reported test can be used outside hospitals and clinical laboratories [57].

Apart from Cas-13, Cas-12 effectors have been also employed for coronavirus diagnosis. For that, CRISPR-Cas12 was combined to DETCTR platform with the DNA endonuclease-targeted CRISPR trans reporter (DETCTR) platform. Extracted RNA is first retro-transcripted to DNA, followed by isothermal amplification using the RT-RPA enzyme. The Cas12a-gRNA complex is then activated through identification of a specific sequence in the amplified DNA, leading to the cleavage of reporters as it was described for the SHERLOCK platform. Based on this principle, different fluorescent lateral flow assays have been developed and tested in clinical samples, showing excellent results [58].

CRISPR-Cas9 effectors have been also exploited to construct a triple line lateral flow assay for the simultaneous dual gene (E and Orf1ab) detection of SARS-Cov2. The assay is based on the recognition between the Cas9/sgRNA and DNA-AuNPs probes. After binding, the formed complex accumulates on the test line and generates a visible colorimetric signal. The originality of this strip is the simultaneous detection of more than one gene which is not available by using Cas13 and 12 requiring nonspecific cleavage [59].

Finally, these studies confirm that CRISPR technology holds a great potential for the diagnosis of the novel coronavirus. It constitutes a promising alternative for the RT-qPCR technique, mainly due to its low cost, simple fabrication and sensitivity.

#### 2.2.2. Aptamer-Based Biosensing

Aptamers are a class of bio-inspired receptors composed of single stranded DNA or RNA. They are selected by a combinatorial process called selection of ligands by exponantial enrichment (SELEX)for their affinity and specificity to a target [60]. They are able to bind their targets with high affinity and specificity comparable to antibodies. In addition, they are characterized by many advantages compared to antibodies, including cheap and simple chemical synthesis as well as stability in temperature and pH variations [61]. Moreover, aptamers can be easily functionalized with various chemical groups and labels allowing their use in different analytical applications [62].

In April 2020, Song et al selected the first aptamer specific for receptor binding domain of the SARS-CoV-2 spike glycoprotein. SELEX was carried out by incubating a ssDNA (single stranded DNA) library comprising 40 random nucleotides with RBD protein immobilized on A-beads. After a total of twelve rounds of selection and amplification, two sequences have been chosen. They exhibited a high affinity for the novel coronavirus with respective Kd values of 5.8 and 19.9 nM. In addition to the experimental competition, molecular dynamic simulations suggested that the selected aptamers bind to identical amino acids of RBD, thus providing a promising issue for coronavirus diagnosis, prevention and treatment [63]. The feasibility of the selected aptamer for spike protein detection has been subsequently, investigated by another research group. For that, the biosensor platform was constructed by immobilizing the amino-modified S-specific aptamer on a silicon thin film transistor. A wide range of potentials was applied to obtain a broader concentration dependency and study the binding mechanism between the aptamer and the spike protein. As compared to the FET-based immunosensor discussed above [64] the FET-based aptasensor showed a similar response concerning for S protein detection; the ratio increased by approximately 10% at the maximum spike concentration compared to the 0% response [64]. Therefore, this study confirms the potential application of the aptamer targeting the S protein in COVID-19 accurate diagnosis.

Apart from the spike protein, Nucleocapsid protein (NP) has been also targeted by SELEX technology. Zhang et al selected four aptamers specific for N protein with high affinity lower than 5 nM.For that, a total of five rounds of selection were performed starting from a library of 76 nucleotides comprising a randomized sequence of 36 nucleotides. After being selected, aptamers were employed in a sandwich-type assay based on four aptamer pairs, which form a ternary complex with NP. Combined with antibodies, this principle was first applied in an enzyme linked immunosorbent assay showing a high sensitivity. Then, an immunochromatographic strip was developed using gold nanoparticles to allow the visual detection of NP in urine and serum [65]. The selected aptamers hold a great promise for aptamer-based diagnosis and treatment of SARS-CoV-2. However, the developed tests require pairs of aptamers and antibodies thus increasing the device cost and complexity.

Besides to aptamers specifically selected for the novel coronavirus, the potential use of SARS-CoV-1 specific aptamers has been also investigated. Based on the fact that the sequence of SARS-CoV-2 N protein exhibits a homology of 91% with that of SARS-CoV, Chen et al studied the binding capacities of the SARS-CoV aptamer to detect the new coronavirus. Enzyme-linked aptamer binding assay (ELAA) demonstrated that the three tested aptamers exhibit a binding affinity for the SARS-CoV-2 N protein opening the way for their application in COVID-19 diagnosis and treatment. Aiming to validate that, the binding efficiency of aptamer 2 was compared to that of a commercial antibody using Western blot showing similar results. These aptamers could thus be a good alternative for antibodies and immunoassays, currently used for coronavirus detection. Moreover, the use of these aptamers should provide a low cost and rapid solution by avoiding the selection of new candidates specific to SARS-CoV-2 [66].

Recently, an interesting study was published by exploring the binding affinity of a nucleocapsid protein specific aptamer and proximity ligation strategy. The principle is based on targeting the antigenic protein with two different aptamers in order to bring the ligation DNA region into close proximity and initiate ligation-dependent qPCR(quantitative PCR) amplification (Figure 2). The protein recognition is thus translated into a detectable qPCR signal allowing sensitive quantification of coronavirus. Ct(Cycle Threshold)value changes were used to monitor the presence and concentration of the virus in serum samples. The proposed technique offers a universal diagnosis platform for coronavirus and other pathogens. Concerning therapy, the authors studied its feasibility for screening and investigation of potential neutralizing aptamers [67].

Despite the excellent characteristics of aptamers, few research reports explored their advantages in COVID-19 diagnosis. Based on the excellent results of the aptasensors discussed above, more techniques will be certainly described in the future by exploring the different sequences selected either for SARS-CoV or SARS-CoV-2.

#### 2.2.3. Antisense Oligonucleotides-Based Biosensing

Based on the predictable and specific hybridization of complementary bases, DNA-RNA hybridization has been widely used in RT-PCR and other biomedical diagnosis techniques. Hybridization is mainly dependent on melting temperature; complementary strands can hybridize with each other when the temperature is slightly lower than their melting temperature, while a single mismatch can cause the melting temperature to decrease significantly [68]. Qui et al. reported a complementary DNA-based biosensor for the selective detection of viral sequences including RdRp, ORF1ab, and E genes from SARS-Cov-2. In this work, dual functional biosensing was undergone by combining the plasmonic photothermal effect (PPT) and localized surface plasmon resonance (LSPR)transduction The principle is based on the hybridization between the thiol-cDNA receptors immobilized on two dimensional gold nanoisland-chip and the target genes. The PPT heat increases in the situ hybridization temperature thus allowing the discrimination of the specific gene in a multigene mixture. On the other hand, LSPR is highly sensitive to local variation, including the refractive index change and molecular binding thus allowingreal-time and label-free monitoring [10]. Electrochemical detection was also applied in the antisense-oligonucleotide-based biosensing of SARS-- genetic material. In this context, Alafeef et al developed a paper-based electrochemical platform based on a ssDNA-capped gold nanoparticles as biosensing element of SARS-CoV-2 RNA. For enhanced analytical performances, four probes were designed to bind two regions within the protein N gene. The specific cDNA (complementary DNA)-viral RNA hybridization induces charge and electron mobility changes on the graphene conductive film thus inducing a potential variation. Based on this principle, the digital electrochemical detection of SARS-CoV-2 RNA was successfully carried out in clinical samples with high sensitivity and selectivity within a short time of 5 min [69].

Owing to their simple synthesis and low cost, antisense oligonucleotides could be more advantageous than the other bioreceptors discussed in this review including aptamers and antibodies.

Table 3 and Table 4 summarize the different strategies based on nucleic acid binding applied for COVID-19 detection.

## 3. Role of Nanotechnologies in COVID-19 Treatment

In recent years, nanotechnology and nanoengineering have shown a remarkable progress. Nanomaterials have been widely studied and applied in numerous fields including medicine, showing great potential in the diagnosis, prevention, control and treatment of many diseases [84,85]. This, is mainly due to their small size, large surface area, optical properties, biocompatibility, solubility, high reactivity and surface functionality [86].

After discussing the different COVID-19 biosensing platforms integrating nanomaterials in the first part of this review, we focus in this part on the different therapeutic strategies based on nanotechnology. Basically, nanoparticles can be used as drug carriers and viral inhibitors. Moreover, they hold a great potential in vaccine research for the novel coronavirus.

### 3.1. Drug and Vaccine Delivery

Nanomaterials have been reported as excellent carriers for delivering drugs, with high selectivity, to targeted locations in organism. It has been demonstrated that nanocarriers reduce therapeutic agent toxicity, immunogenicity and side effects while increasing drug efficacy. Moreover, certain nanomaterials have been already approved by the FDA for drug delivery such as polylactic-co-glycolic-acid and lipid nanoparticle systems [87]. 

In vaccine design, nanocarriers constitute a key element either in antigen encapsulation and presentation or boosting of the immune system via the co-delivery of antigen and adjuvant [88]. The ability of nanoparticles to deliver genomic materials including mRNA (messenger RNA) and DNA has been explored in developing vaccine development. For instance, the ZIKA vaccine was developed by encapsulating the viral mRNA in lipid nanoparticles (LPNs) [89]. The same strategy was explored by Pfizer and Moderna in the development of the COVID-19 vaccine [90]. mRNA vaccines offer many advantages includingrapid and inexpensive development in addition to the low risk of integration with the host genome. The principle is based on mRNA in vitro transcription from the DNA template and then the corresponding spike protein is translated and expressed within the host cell thusreducing the side effects. In parallel, by using lipid nanoparticles formulation, the genetic information is precisely delivered with an adjuvant effect to antigen-presenting cells [91]. In addition to LPN delivery systems, other nanocarriers could be employed in vaccine development. For instance, it has been demonstrated thattheNuvec system, based on silica nanoparticles, could provide more safety; it does not disrupt the membrane and does not induce an inflammatory response. Being in the last stage of clinical trials, this platform holds a great potential as an efficient and safe delivery system for DNA/RNA vaccines and may be useful in combating the COVID-19 pandemic [92].

Besides to vaccine delivery, nanomaterials could be used as antiviral carriers for SARS-CoV-2 treatment. For instance, Ullah et al. proposed a siRNA-based antiviral strategy by using cationic liposomes as a delivery system. Aiming to target ACE2 receptors, the authors proposed an aerosol formulation for intranasal or intratracheal administration. This targeting can be achieved by functionalizing lipid/polymer nanocarriers with highly specific receptors such as antibodies or aptamers. Such a strategy could promote the use of siRNA (small inhibitory RNA)for silencing and knockdown of viral genes [93].

### 3.2. Antivirals

Nanomaterials can be successfully conjugated to various functional groups, linkers and bioactive molecules. Moreover, it has been demonstrated that nanoparticles are able to cross the blood-brain and blood-air barriers as well as lymphatic system [94,95]. Furthermore, nanomaterials can be specifically used to target an organ, cellular or intercellular site involved in the viral infection [96]. Owing to these features, several nanoparticles (silver, gold, TiO_2_, SiO_2_…) have been used as antivirals against different viruses including hepatitis B, H3N1 and H1N1viruses [95]. In general, nanomaterials act by inactivating the virus, preventing its binding to the target cell or blocking its replication [97]. For instance, viral penetration can be blocked by exploring the ability of certain nanomaterials to mimic heparan-sulfate proteoglycans (HSPG), the highly conserved target of viral attachment ligands. Based on this principle, Cogno et al designed antiviral nanoparticles with long and flexible linkers mimicking HSPG, allowing the effective viral association which leads to irreversible viral deformation. The authors confirmed that these particles present a strong antiviral activity against herpes simplex virus (HSV), human papilloma virus, respiratory syncytial virus (RSV), dengue and lenti virus [98].

Based on these findings, nanomaterials-based antiviral strategy could be used as promising tool for SARS-CoV-2 inhibition. In the literature, different reviews have discussed the potential of nanotechnology in combating COVID-19 infection [87,99]. In their interesting review, Gurunathan et al proposed a hypothetical antiviral mechanism based on ACE2-coated nanoflowers and ACE2-quantum dots. These conjugates could block the SARS-CoV-2 entry by inhibiting the viral attachment to ACE2 receptors on the host cell. Moreover, they could inhibit the activation of the accessory serine protease TMPRSS2 [94]. In another study based on mimicking the host cell surface, Zhang et al designed cellular nanosponges for COVID-19 neutralization. Made of plasma membranes derived from human lung epithelial type II cells or human macrophages, these nanosponges display an ACE2 receptor thus allowing the blockage of viral entry [99]. In brief, great path could be opened, by these promising results, of anti-COVID 19 nanotechnology-based approaches.

## 4. Conclusions

The health crisis caused by the COVID-19 pandemic showed the key role of biotechnologies in biomedical research. As COVID-19 patients display nonspecific symptoms, efficient diagnostic approaches are highly required. Many efforts have been devoted worldwide to meet the urgent need for cost-effective mass screening. In this context, a large array of biosensors has been developed based on different bioreceptors and detection techniques. Many research groups have investigated immunochemical techniques based on antibodies, peptides and aptamers, combined with electrochemical and optical transducers for the sensitive and specific detection of viral antigens and/or immunoglobulins. In parallel, viral genes have also been targeted by using different types of nucleic acids including CRISPR and antisense oligonucleotides. Most part of the developed biosensors was portable allowing rapid, cheap and easy to use point of care testing of SARS-CoV-2. Moreover, some techniques allow the sensitive determination of more than one indicator providing a complete diagnosis and monitoring platform. Furthermore, dual platforms combining simultaneous sample collection and antigen sensing have also been reported. Finally, the selectivity of these technologies has been demonstrated against different types of human pathogens. 

Besides diagnosis, vaccination is also of a great importance in preventing the propagation of viruses, in particular, with the fast spreading UK virus variant which is raising alarms [99]. In this review, we focused on the role of nanotechnology in the design of SARS-CoV-2 vaccines and adjuvants. Moderna and Pfizer, the frontrunners among COVID-19 vaccine developers, employed nanoparticles in vaccine delivery. In this context, the genomic material (mRNA) is principally delivered via lipid nanoparticles previously approved by the FDA. LPNs provide a specific targeting and prevent systemic and nuclease degradation of the vaccine. Finally, the use of nanomaterials as antivirals was also discussed in this review. Despite the promising results of nanoparticle-based antivirals applied in fighting various viruses, few reports have been published about COVID-19. These nanoscale entities thus hold a great challenge for researchers to overcome limitations of the traditional delivery systems and antivirals.

## Figures and Tables

**Figure 1 sensors-21-01485-f001:**
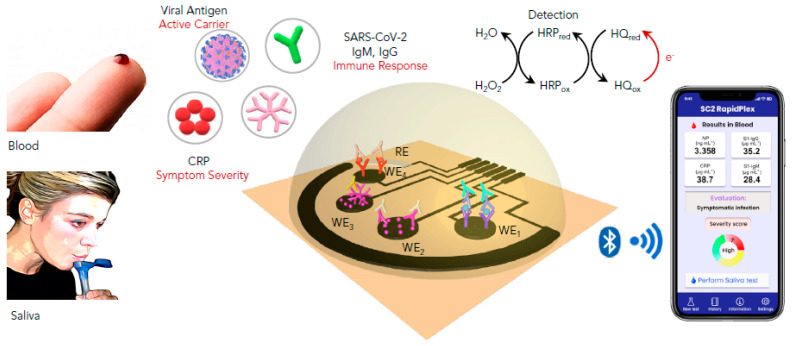
Schematic representation of the graphene-based electrochemical telemedicine platform for multiplexed detection of CRP, (SARS-CoV2) antigens, IgG (Immunoglobulin G) and IgM (Immunoglobulin M). This RapidPlex is composed of four graphene working electrodes (WEs); Ag/AgCl reference electrode (RE), and a graphene counter electrode (CE). All of them patterned on a polyimide (PI) substrate via CO2 laser engraving, and a fast, high-throughput. Data can be wirelessly transmitted to a mobile user interface.Reprinted with permission from [20].

**Figure 2 sensors-21-01485-f002:**
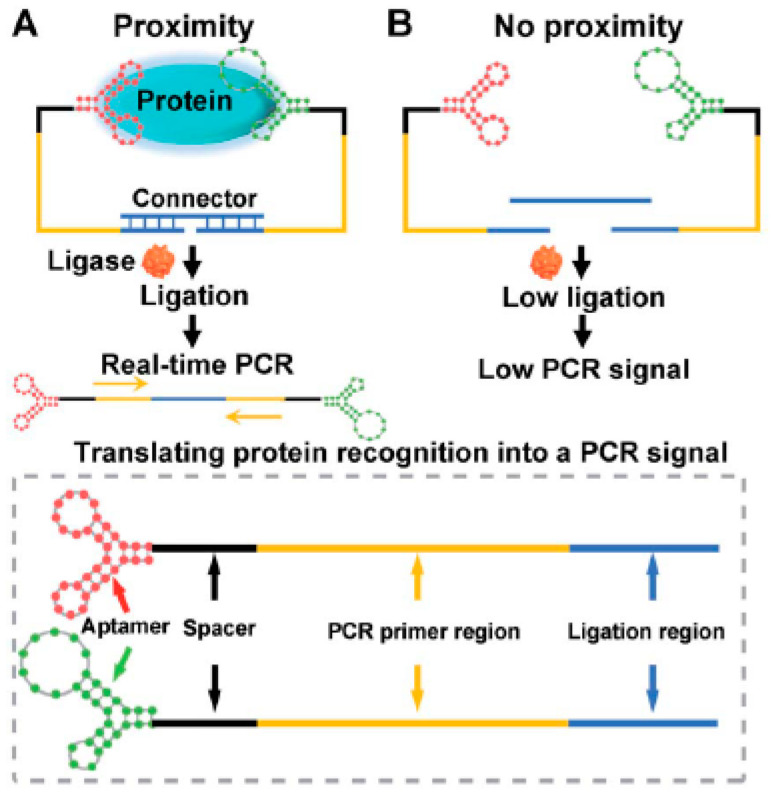
Schematic representation of the aptamer-assisted proximity ligation assay for COVID-19 antigens. The aptamer- proximity ligation assay has two important components: two proximity ligation probes: an aptamer region for target recognition and a spacer region to minimize the structural steric hindrance(obstruction). (**A**)The binding of the two aptamers to the same protein target could bring the ligation region into close proximity. Then, the two ligated aptamers hybridize with the connector template, to form a DNA (Desoxyribonucleic acid) complex used for subsequent ligation-dependent qPCR (quantitative PCR) amplification. (**B**)The absence of the target hampers the proximity of the two probes leading to a low ligation event and weak nonspecific qPCR signals. Reprinted with permission from [68].

**Table 1 sensors-21-01485-t001:** Electrochemical biosensors for SARS-CoV-2 detection.

Detectionmethod	Target Genes	LOD	Time	Portability	Ref
Electricalconductivitychanges	S protein	2.25 × 10^−6^ μg/mL	10–30 s	Portable	[36]
FET	S protein	10^−9^ μg/mL	Real time	Nonportable	[11]
Gr-FET	S1 protein (RBD)	<1.538 × 10^−4^ μg/mL	2 min	Nonportable	[23]
Differential pulse voltammetry (DPV)	S protein / N protein	Sprotein: 19 × 10^−3^ μg/mL Nprotein: 8 × 10^−3^ μg/mL	30 min	Portable	[18]
Impedancespectroscopy (EIS)	SARS-CoV-2 antibodies	Not available	<5 min	Portable	[37]
DPV	SARS-CoV-2 S-glycoproteins	1.68 × 10^−22^ μg/mL	1 min	Nonportable	[38]
DPV+ OCP-EIS	S1-IgG, S1-IgM, NP, CRP	Not available	1 min	Portable	[20]
Square-wavevoltammetry	SARS-CoV-2 antigen	0.8 pg/mL	Unknown	Nonportable	[21]

FET: Field Effect Transistor, RBD (Receptor Binding Domain), CRP (C-Reactive Protein).

**Table 2 sensors-21-01485-t002:** Optical biosensors for SARS-CoV-2 detection.

Detectionmethod	Target Genes	LOD	Time	Portability	Ref
Surface plasmonresonance (SPR)	Nucleocapsidof anti-SARS-CoV-2 antibodies	Not available	15 min	Portable	[39]
LFIA	IgM and IgGantibodies	Not available	15 min	Portable	[31]
LFIA	SARS-CoV-2 NP antigen	250 pfu/µL	20 min	Portable	[40]
LFIA	S1 protein	1.86 × 10^5^ copies/mL	20 min	Portable	[41]
Colloidal gold-nanoparticle-basedLFIA	IgM antibodies	Not available	15 min	Portable	[42]
SERS-LFIA	Anti-SARS-CoV-2 IgM/IgG	Not available	Real time	Portable	[43]
ELISA+GICA	IgM and IgG	Not available	GICA: 10min	Nonportable	[44]
Grating-coupledfluorescentplasmonic	Antibodiesagainst RBD; spike S1 fragment; spike S1 S2 extracellulardomain, andNprotein	Not available	<30 min	Nonportable	[45]
Immunochromatographicstrip	IgM orIgGantibody	Not available	15 min	Portable	[46]
Fluorescenceimmunochromatographic	N protein	Not available	10 min	Portable	[35]
Nanoplasmonicresonancesensor	S protein	0.37 copies/µL	<15 min	Portable	[47]
Microfluidicimmunoassay-basedfluorescence	IgG/IgM/Antigen of SARS-CoV-2	Not available	<15 min	Portable	[48]
Peptide-basedluminescentimmunoassay	IgG and IgM	Not available	Unknown	Nonportable	[30]
rN-based and rS-basedELISAs	IgM and IgGantibodies	Not available	Unknown	Nonportable	[49]
RT-LAMP-NBS	F1ab and N protein genes	12 copies/µL	≈1 h	Portable	[13]
AuNR-based SPR	S protein	Not available	Fewminutes	Nonportable	[29]
LFIA	Anti-SARS-CoV-2 IgA	Not available	15 min	Portable	[34]

PFU (plaque-forming unit): a measure used invirologyto describe the number of virus particles capable of forming plaques per unit volume, LFIA (Lateral Flow immunoassay), ELISA (Enzyme Linked Immunosorbent Assay), IgG (Immunoglobulin G), IgM (Immunoglobulin M), NP (Nucleocapsid protein).

**Table 3 sensors-21-01485-t003:** Nucleic acid-based-biosensors for SARS-CoV-2 detection.

Type	Detectionmethod	Target Genes	LOD	Time	Portability	Ref
CRISPR	All-In-One Dual CRISPR-Cas12a	SARS-CoV-2 N RNA	4.6 copies/µL	40 min	Nonportable	[70]
	CRISPR-Cas12 based LFA + RT-LAMP	E and N protein genes	10 copies/μL	<40 min	Portable	[71]
	CRISPR-Cas9-based, LFA	E and Orf1ab genes	4 copies/µL	<1 h	Nonportable	[59]
	CRISPR/dCas9	SARS-CoV-2 N1, N2, and N3 genes	Not available	90 min	Portable	[72]
	CRISPR-Cas13 based lateral flowreadout (SHERLOCK platform)	S gene	0.5 copies/µL	35–70 min	Portable	[56]
	CRISPR-Cas13	ORF genes	106 copies/μL	10 min	Nonportable	[57]
	CRISPR Cas12a/gRNAbased RPA	ORF1ab gene + the N proteingene	4 × 10^−1^ copies/µL	≈50 min	Nonportable	[73]
	AapCas12bbased LAMP (SHERLOCK platform)	N gene	2 copies/µL	<80 min	Nonportable	[53]
	CRISPR-Cas13 based lateral flow	S and ORF1ab protein genes	10 copies/µL	<1h	Nonportable	[54]
	CRISPR-Cas13-nCoV	ORF1ab genes	3 copies/µL	40 min	Nonportable	[74]
	RPA-CRISPR-Cas12+ lateral flowsystem	ORF1ab genes	10 copies/μL	10 min	Portable	[58]
Antisense oligonucleotides	PPTeffect + localized SPR	RdRp, ORF1ab, and E genes	113 × 10^3^ copies/µL	Real time	Nonportable	[10]
	DNAnanoswitchbasedNASBA	SARS-CoV-2 RNA	≈100 copies/µL	2 h	Nonportable	[75]
	LAMP + RAMP	ORF1ab	LAMP: 1 copies/µL RAMP: 10^−1^ copies/µL	Real time	Nonportable	[76]
	RT-LAMP	ORF1ab + S gene	ORF1ab gene: 0.8 copies/µL S gene: 8 copies/µL	60 min	Nonportable	[77]
	RT-LAMP	ORF1a/b, S and N genes	8 × 10^−2^ copies/µL	30 min	Nonportable	[78]
	RT-LAMP	Nsp3 gene	6.66 copies/µL	30 min	Nonportable	[79]
	RT-LAMP	nucleotide 2941-3420 ofthe COVID-19 completegenome (MN908947)	Not available	<30 min	Portable	[80]
	RT-LAMP	ORF1ab gene	10 copies/µL	15–40 min	Nonportable	[81]
	Graphene-basedelectrochemical+ Au-NPs	N gene	6.9 copies/μL	<5 min	Portable	[69]
	Plasmoniceffectbasedcolorimetricbiosensing (Au-NPs)	N gene	40 × 10^−2^ copies/µL	≈10 min	Nonportable	[82]
	Isothermalrollingcircleamplification	N + S genes	10^−3^ copy/µL	<2 h	Nonportable	[83]

LFA: Lateral flow Assay, RT-LAMP (Reverse transcription loop-mediated isothermal amplification), CRISPR (Clustered Regularly Interspaced Short Palindromic Repeats), gRNA (Guide ribonucleic acid), RPA (recombinase polymerase isothermal amplification), PPT (photothermal effect), DNA (Desoxyribonucleic acid), NASBA (Nucleic acid sequence-based amplification), RAMP (repeat-associated mysterious proteins).

**Table 4 sensors-21-01485-t004:** Aptamer-based-biosensors for SARS-CoV-2 detection.

Detectionmethod	Target Genes	LOD	Time	Portability	Ref
Intrinsicsiliconthin film transistor (FET)	S protein	0.75 ng/mL	Unknown	Nonportable	[64]
ELISA + colloidalgoldimmuno-chromatographicstrips	N protein	1 ng/mL	15 min	Nonportable	[65]
ELAA	N protein	10 ng/mL	Unknown	Nonportable	[66]
aptamer-assistedproximityligationassay	serumnucleocapsidprotein	37.5 ng/ mL	2 h	Nonportable	[67]

ELISA (Enzyme linked Immunosorbent Assay), ELAA (Enzyme Linked Aptamer Assay).

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
