# Peer review of "Analysis of Recent Bio-/Nanotechnologies for Coronavirus Diagnosis and Therapy"

_sensors, 2021, doi:10.3390/s21041485_

Round 1

Reviewer 1 Report

Specific comments:

  1. Please give a more detailed description of figures 1 and 2, so that readers can better understand the content of the article.
  2. Tables 1 and 2 should adopt a unified unit as far as possible, so as to compare the detection limit and detection time of different technical means more clearly.

Author Response

  1. Please give a more detailed description of figures 1 and 2, so that readers can better understand the content of the article.

More detailed description was added to the figures.

  1. Tables 1 and 2 should adopt a unified unit as far as possible, so as to compare the detection limit and detection time of different technical means more clearly

The tables have been unified, for that two tables were added.

Reviewer 2 Report

The authors have done a good job in describing new nanotechnologies for diagnosing and enhancing treatments for coronavirus. The paper is fairly well written, but has some grammatical and sentence structures that could be improved for maintaining a better or more formal writing style for conventional science reporting. The following suggestions are provided to give some guidance for improving the manuscript presentation for achieving easier reading and grammatical structure.

Please use direct statements for first-topic sentences, i.e., those beginning new paragraphs, instead of using indirect statements such as ones with independent clauses, prepositions, adverbs etc., followed by a comma. Indirect statements in introductory sentences to paragraph are generally regarded as poor writing style for good grammatical presentations.

Please remove back-slashes (/) in Tables 1 and 2, replacing these with actual useful information (such as the following suggestions):

Table 1 - column LOG - unknown or give estimated range or na (not available); column Portability - nonportable 

Table 2 - column Portability - nonportable

References: [a major revision in formatting needed here!]

All journal citations need to be changed from AMA format to MDPI format (with no issues in parentheses, colons, or p.)  

Author Response

  1. Please use direct statements for first-topic sentences, i.e., those beginning new paragraphs, instead of using indirect statements such as ones with independent clauses, prepositions, adverbs etc., followed by a comma. Indirect statements in introductory sentences to paragraph are generally regarded as poor writing style for good grammatical presentations.

Some paragraphs have been modified, for example; the paragraph in line 38, 56 and 140.

  1. Please remove back-slashes (/) in Tables 1 and 2, replacing these with actual useful information (such as the following suggestions):

Done.

  1. Table 1 - column LOG - unknown or give estimated range or na (not available); column Portability - nonportable 

Done

  1. Table 2 - column Portability – nonportable

Done

  1. References: [a major revision in formatting needed here!] All journal citations need to be changed from AMA format to MDPI format (with no issues in parentheses, colons, or p.)  

Journal citations were changed to MDPI format.

Reviewer 3 Report

The paper by Marty and collaborators can be considered a minireview about the biosensors and nanotechnological methods recently introduced as diagnostic and therapeutic tools to fight the novel coronavirus disease (COVID-19). The manuscript offers to the reader a large perspective of the recent advances, covering a rich variety of methods from immunochemical to nucleic acid-based biosensors. The paper is well organised and language is fluent. It represents a good landmark for future research in this emerging field.

  1. In Table 1, in the first line: the definition ’Target genes’ is not appropriate for protein and antibody.
  2. Data in Tables 1 and 2 should be unified in terms of LOD (molarity or mass/volume) for a better comparison among the techniques.
  3. Column format in the tables should be re-adjusted (the text should not be justified and words should be written entirely in one line when possible)

Author Response

  1. In Table 1, in the first line: the definition ’Target genes’ is not appropriate for protein and antibody.

It has been modified to targets, and the term “target genes” was kept just for nucleic acid based methods.

2.  Data in Tables 1 and 2 should be unified in terms of LOD (molarity or mass/volume) for a better comparison among the techniques.

The tables have been unified, for that two tables were added.

3.  Column format in the tables should be re-adjusted (the text should not be justified and words should be written entirely in one line when possible)

Column format was adjusted.

Round 2

Reviewer 2 Report

The authors have done a diligent job in correcting the major limitations and required corrections of the manuscript.